# The Threshold Effect of Swine Epidemics on the Pig Supply in China

**DOI:** 10.3390/ani12192595

**Published:** 2022-09-28

**Authors:** Yunfei Jia, Wenshan Sun, Guifang Su, Junguo Hua, Zejun He

**Affiliations:** College of Economic and Management, Henan Agricultural University, Zhengzhou 450002, China

**Keywords:** swine epidemics, pig supply, threshold effect, pig industry

## Abstract

**Simple Summary:**

China is the world’s largest pork producer and consumer, and pork is among the most important foods for people. Therefore, pig supply has a bearing on residents’ quality of life and the security of the pig industry. However, swine epidemics such as African swine fever have led to drastic changes in pig supply. This study analyzed the mechanism of the effect of swine epidemics on nonlinear shocks to pig supply, and monthly data on pig supply from January 2012 to June 2020 were applied to study the threshold effect of swine epidemics on pig stock and slaughter in China empirically. Our results contribute to devising relevant measures for the prevention and control of swine epidemics.

**Abstract:**

The pig industry is the pillar industry of animal husbandry in China, and epidemics can lead to drastic changes in pig supply, affecting the healthy development of the pig industry and residents’ quality of life. This study analyzed the mechanism of the effect of swine epidemics on nonlinear shocks to pig supply, and monthly data on pig supply from January 2012 to June 2020 were applied to study the threshold effect of swine epidemics on pig stock and slaughter in China empirically, using the index of swine epidemics’ width (ISEW) as the threshold variable. The results of this study were as follows: (1) The influence of the ISEW over 7 months on pig stock in China was divided into two ranges, and the pig stock did not change significantly when the ISEW was less than 0.25. Swine epidemics had a significantly negative impact on the pig stock when the ISEW was larger than 0.25. (2) The influence of the ISEW over 8 months on pig slaughter was also divided into two ranges. When the ISEW was less than 0.33, epidemics had a positive and significant effect on pig slaughter, while epidemics had a marked negative impact on pig slaughter when the ISEW was greater than 0.33. Based on these conclusions, this study proposed relevant measures for the prevention and control of swine epidemics.

## 1. Introduction

Pork is a vital commodity in the food system of China [1], and pork production accounts for more than 55% of the total meat production in China. Abnormal fluctuations in the pig supply not only provoke variations in pig prices, causing loses to both agribusinesses and consumers, but also lead to an imbalance in national economic development and output–input asymmetry. The healthy development of the pig industry is closely related to food safety and the steady growth of the national economy in China. Fluctuation and regulation of the pig supply are among the major risks and problems faced by China’s pig industry. Swine epidemics, as external shocks with propagation, have a wide range of influence and a large degree of impact. Swine epidemics are the main cause of uncertain changes in the pig supply, which can cause drastic shocks to the pig supply and the hog industry in China. For example, the outbreak and rapid spread of African swine fever in China in 2018, caused severe fluctuations in the pig supply [2]. Compared with 2018, in 2019, the live pig stock decreased by 27.5%, and the number of pigs slaughtered decreased by 21.6%, resulting in a serious setback in Chinese pig production.

Overlapping the outbreak of COVID-2019 continues to spread globally, uncertainty about pork imports and exports has increased, and supply chains in China’s pig market have been battered. During the outbreak of the swine epidemic, a shortage of live pigs led to a sharp rise in pork prices. Under the guidance of a price signal, a large number of breeders take advantage of the favorable opportunity of the market situation to supplement and re-raise, resulting in the oversupply of pigs, which may induce a price collapse in the post-epidemic era, affect the consumer price index (CPI) [3] and deepen the impact on the supply of pigs and the stability of the pig market.

In the context of the frequent occurrence of various uncertain factors influencing pig supply, this study focuses on the swine epidemic, which is a direct and important factor that has a great impact on pig supply, studies the conversion mechanism of abnormal fluctuations in pig supply that are caused by swine epidemics, and identifies the early warning signals in swine epidemics that cause drastic fluctuations in pig supply. This study is helpful to alleviate the impact of swine epidemics on pig supply, and is of great significance to maintain the healthy development of the pig industry in China.

The outbreak of swine epidemics is sudden and widespread in nature, which is an important uncertainty shock to the hog industry, proving a difficult problem in balancing the relationship between supply and demand for pork.

On the supply side, swine epidemics will cause an increase in the morbidity and mortality of pigs, which will have a negative effect on slaughter. Additionally, swine epidemics usually infect the susceptible herd of sows and piglets first, which will bring about abortion easily in sows [4] and directly slow down the recovery of pig production capacity, which will seriously affect the pig supply in the next phase. The government, in order to limit the rapidly development of the epidemic, introduced control policies including measures such as the culling of susceptible animals, cleaning and disinfecting infected premises, and imposing restrictions on pig transports in surveillance and protection zones, which are established around the infected premises [5,6], leading to a rise in the number of pigs culled, restriction of pig transportation, a further reduction in pig supply, significant expansion of the supply–demand gap, and a general increase in pig prices nationwide. Together with the cyclical factors of pig supply, the time lagging of its production leads to the fact that pig supply will not be restored within a short period of time, so the impact of swine epidemics on pig supply is quite remarkable.

On the demand side, consumers are sensitive to food safety problems such as the outbreak of animal diseases [7,8], so the outbreak of swine epidemics directly causes great panic in pig consumers, having a huge impact on consumer demand for food. Therefore, at the beginning of the outbreak, consumers tend to choose other substitutes of pork, such as chicken and beef [9], reducing the demand for pork, which directly affects the consumption of pork in the current period and causes the price of pork to fall. However, it is difficult to change consumption preferences in the short term, as rigid demand for pork cannot be replaced quickly [10]; moreover, consumer confidence in pork is elastic, and along with increased publicity and guidance by the government, consumer confidence in pork consumption has been rebuilt [11], increasing the demand for pork, but it is difficult to recover pig supply in the short term, resulting in the expansion of the supply and demand gap. As a result, the price of pork during the outbreak phase of swine epidemics rises with the increase in consumer confidence in pork [12,13]. Overall, swine epidemics also have a significant impact on pork prices, and the impact on pork price can last a long time, and weaken the role of the self-regulatory mechanism of pork market prices, creating and imbalance between supply and demand for pork [14], so effective regulation is needed with the help of government policy instruments.

Previous research concluded that swine epidemics can lead to fluctuations in the pig supply by affecting supply and demand [15,16], but with little concern for the nonlinear fluctuation (nonlinear fluctuation means the occurrence of non-proportional, that is, different degrees of swine epidemics impact the pig supply with differences) of the pig supply caused by swine epidemics and their mechanisms. In fact, as affected by major swine epidemics, the change in pig supply is not only influenced by the epidemic itself, but also by the producers’ decision making regarding production, based on their price expectations and the peer effect between them, which would result in the Matthew effect [17], which is a phenomenon presented by Merton: “for to everyone who has, more will be given and he will grow richer; but from the one who has not, even what he has will be taken away” and in this study, the Matthew effect is embodied by the higher pork prices leading to the rising prices of hogs, so farmers choose to extend the feeding time of hogs, resulting in a further reduction in the supply of pigs. At this time, the hog market is in excess demand over supply, hog prices and pork prices will rise further and the rate of increase will become larger (i.e., nonlinear fluctuations are produced), making the normal mechanism of price regulation a problem.

Therefore, in the present study, the mechanism of the fluctuation in the pig supply caused by swine epidemics was explored from a nonlinear perspective, using a threshold regression model to measure the impact of swine epidemics on the live pig stock and the number of slaughtered pigs in China under different ranges. Our aim was to identify the threshold value of the epidemic index that leads to the fluctuation in the pig supply, which would provide a theoretical basis and support for relevant departments of the government to implement policies to stabilize the supply under the impact of major pig epidemics.

## 2. Microscopic Mechanism Analysis

### 2.1. The Conduction Path and the Three Phases of the Impact of Swine Epidemics on the Pig Supply

As shown in Figure 1, swine epidemics trigger considerable shocks to the hog industry by influencing the pig supply from two aspects. On the one hand, the direct impact of epidemics on the pig supply lies in the supply side [18]. The outbreak and the spread of the epidemic will inevitably cause the death or culling of sick pigs, resulting in a huge gap between the supply of pigs and the demand for pork, which has poor elasticity, and a sharp decline in the number of sows and piglets, causing a slow recovery of pig supply capacity [19]. When a major epidemic occurs, the cost of preventing the spread of the epidemic continues to increase, which seriously restricts the pig supply and the operation efficiency of the industrial chain of small breeding entities in the epidemic area [20,21]. In addition, due to the superposition of the restrictions on pig transportation policies and epidemic prevention and controlling policies, the recovery speed and quality of pig supply will slow down. On the other hand, the indirect impact of epidemics on the supply of pig lies in changing farmers’ behavior during pig production [22]. In order to avoid health risks, consumers’ demand for pork has reduced with the spread of information about the epidemics by the news media in the short term, which will decrease the price of pork. The high mortality and low productivity of pigs [23] and the temporary low demand for pork [1,24], caused by the outbreak of swine epidemics, have brought about an increase in sunk costs for most pig farmers, resulting in a funding gap. Due to the unclear market expectations, farmers choose to maintain a prudent attitude in decision making, and lack the enthusiasm for pig re-breeding and supplementary hurdles in order to avoid risks, affecting the supply of pigs indirectly.

At different time points after the outbreak, the different degrees of the severity of the epidemic would have different effects on the pig stock and the slaughter rate, resulting in an imbalance in pig supply. Swine epidemics usually go through three phases: the initial phase, the explosive phase and the recovery phase. During these phases, the supply and demand sides of pork are affected to different degrees, and how the farmers react varies.

(1) During the initial phase, the areas affected by the epidemic are small and pig mortality is low. The state introduces timely measures to prevent and control the epidemics, including culling sick pigs in infected areas [25], leading to a reduction in the pig stock and the number of pigs slaughtered, with a light impact.

(2) The explosive phase is a critical and risky phase that has severe impacts on the pig industry and the pork market. During the explosive phase, the affected area is further expanded and the number of sick and dead pigs increases markedly owing to the contagious effects of the epidemic. At the same time, farmers choose to lengthen the time till pig slaughter because of the rise in pig prices, which further reduces the pig supply, leading to a rise in pork prices and tightening of the pig supply, resulting in the appearance of the “Matthew effect”, thus the reduction in the pig supply becomes more severe.

(3) When the epidemic is fully controlled and enters the recovery phase, the supply to the pig market will recover slowly according to the recovery rate of the industry chain’s supply capacity and the expansion of new productivity, which might take a long time.

### 2.2. The Micro-Mechanisms of Changes in the Pig Supply

The different severity of swine epidemics and the different phases of swine epidemics have different impacts on pig supply. In general, in the early phase of swine epidemics or in the case of a minor epidemic, the impact on pig supply is shorter and can recover soon, and the social impact of fluctuations in pig supply is comparatively small. When the epidemic is fully controlled and enters the recovery phase, the supply capacity of the pig industry recovers gradually. At the same time, the high price of pigs during the outbreak of swine epidemics leads to a substantial expansion of the capacity of new production in the recovery phase, and the supply of pigs gradually increases or may even reach a surplus. Since the outbreak phase is a critical period that has a great effect on the pig industry and the pork market, this paper focuses on the analysis of the mechanism of the nonlinear impact of swine epidemics on the pig supply during the outbreak phase.

When the swine epidemic is in the outbreak phase, on the demand side, with the measures of controlling the epidemic coming into force and the strength of the propaganda and guidance by the relevant departments, consumer confidence is rebuilt, and the demand for pork rapidly increases. On the supply side, due to the highly infectious effects of the epidemic, areas that have a lot of infected pigs further expand and there is a substantial increase in sick and dead pigs; at the same time, the rise in the pork price leads to rising prices of hogs, and farmers will choose to extend the time for breeding, which further reduces the supply of pigs. Next, this paper analyzes the microscopic mechanism of the nonlinear change in pig supply caused by changes in decision making by farmers under the impact of swine epidemics.

According to the logistic growth model, which is used to describe the growth law, combined with the short-term production function of economics, we expressed the growth law of live pigs near the standard output of live pigs as follows.

At a certain level of technology, the supply of pigs is fully consistent with the short-run production function of a single variable factor according to the law of diminishing marginal returns. With the increase in variable input factors, such as feed consumption, labor, and the prolongation of breeding time, the weight gain of pigs shows an inverted U-shaped trend. As shown in Figure 2A, this study mainly analyzed the breeding time after t_a_, that is, the marginal weight gain of growing pigs decreases with increased breeding time, to determine the optimal time for pig slaughter. To facilitate the analysis, Figure 2B was further used to represent the relationship between breeding time (after t_a_) and pig weight. Although the weight gain of pigs brought about by the lengthened breeding time at this stage decreased, the total pig weight continuously increased, and only the rate of increase gradually slowed.

Breeding time is denoted by “t” and “w” denotes pig weight. t_a_ denotes the point when the marginal weight gain of pigs reaches its maximum; t_0_ denotes the point at which farmers slaughter the pigs to maximize profit without the influence of external shocks.

In this study, we assumed that if the internal factors of pig breeding, including the piglet price, feed price, labor price, and sales territory, remain unchanged throughout pig breeding, and the internal factors are not affected by external shocks, then the appropriate weight of slaughtered pigs will not change and farmers will not make decisions to change the fattening period. However, when affected by the impact of swine epidemics, the supply of pigs will change and the stock and slaughter will decline. The elasticity of consumers’ demand for pork is small and the sensitivity of pork demand to their own changes in price is lower, thus the long-term demand tends to be stable, causing an imbalance in supply and demand in the pig market and a surge in pig prices, which would result in a sharp rise in the profits of breeding pigs, and increase the suitable slaughter weight of fattening pigs. To maximize profit, farmers are inclined to lengthen the breeding period and postpone slaughtering. Thus, the optimal slaughtering time will be extended with the expected price increase.

Due to the impact of swine epidemics, the mortality rate of pigs has increased and the stock has decreased, the elasticity of pig supply is significantly higher than the elasticity of demand. Under the condition that the consumer preference of pork is basically stable, the demand for pork remains stable in the long term. Therefore, farmers expect that pig supply will exceed the demand and the hog price and pork price will rise, and the marginal profit of breeding hog will also increase. In this case, farmers will extend the breeding time, and the optimal breeding time will also be extended with the expected price increase. In this stage, the marginal weight gain, i.e., the daily weight gain of pigs, gradually decreases with the increase in the input of each variable factor and breeding time. The relationship is described by the following equation:(1)$$MP=k\frac{1}{e^{t/t_{0}-1}}$$
where $\frac{1}{e^{t/t_{0}-1}}$ means the change in weight gain efficiency with the increase in breeding time (*t*) during the lengthened fattening period. Assuming *t* = *t*_0_, which means that there is no impact of external factors, such as swine epidemics, the efficiency of weight gain is 1 and the profit margin is 0. At this time, $\frac{1}{e^{t/t_{0}-1}}\in \left[0,1\right]$, because of *t* ≥ *t*_0_. Therefore, the total weight of pigs increased with the lengthening of the fattening period, and the pig’s weight gain function can be expressed as: (2)$$W=W_{0}+\int_{t_{0}}^{t}c\frac{1}{e^{t/t_{0}-1}}dt$$
where *W* is the suitable weight for slaughtering pigs, *W*_0_ is the suitable weight for slaughtering pigs without the impact of swine epidemics, c is the standard daily weight gain of pigs, *t* is the time of fattening pigs, and *t*_0_ is the suitable time to slaughter the pigs without the impact of swine epidemics. *c* = 1, when *t* = *t*_0_; therefore, the daily weight gain of live pigs decreased with the efficiency of weight gain during extension of the fattening period.

To maximize profits, the objective function of farmers’ decision making regarding the time of fattening period is: (3)$$\pi =TR-TC=P\cdot \left(W_{0}+\int_{t_{0}}^{t}c\frac{1}{e^{t/t_{0}-1}}dt\right)-FC-VC\cdot t$$
where *π* means the breeding profit per pig after fattening and slaughtering, *TR* is the total income of each pig, *TC* is the total cost of each pig during the fattening period, *P* is the pig price after the outbreak of a swine epidemic, *FC* is the fixed cost of each pig in the fattening period, and *VC* is the variable cost of each pig during the fattening period, including feed cost, labor cost, cost of epidemic prevention, water and electricity cost. So:(4)$$d\pi =P\cdot c\frac{1}{e^{t/t_{0}-1}}-VC$$

When *dπ* = 0, the objective function has a maximum value, i.e., to maximize the profit of breeding. At this time:(5)$$VC=P\cdot c\frac{1}{e^{t/t_{0}-1}}$$

Given the constant variable cost of daily consumption per pig during the fattening period and the goal of maximizing breeding profits, the higher the price of pigs, the more farmers will choose to lengthen the fattening period of pigs. According to Equation (5), the coefficient of variation for the delayed fattening period is:(6)$$\frac{t}{t_{0}}=1+\ln\frac{P\cdot c}{VC}$$

The gradual increase in the depth and width of the influence of swine epidemics increases the negative impact on the supply of pigs gradually, and the stock and slaughter of pigs decrease. The demand exceeds supply in the current market under the impact of the swine epidemic, which will raise the price of pigs. Under conditions in which the prices of other variable factors stay unchanged, the higher the prices of pigs, the higher the farmers’ expectations for the future price of pigs. Therefore, to maximize the profit of breeding pigs, farmers will be more inclined to choose to lengthen the fattening period, thereby increasing the coefficient of variation of the delayed fattening period, decreasing the amount of slaughter gradually, and tightening the supply of pigs.

The number of slaughtered pigs decreases with the severity of swine epidemics in China. The index of swine epidemics’ width (ISEW) is a comprehensive index sequence obtained by monitoring the situation of swine epidemic diseases in China (mainly including blue ear disease, swine fever, acute diarrhea, high fever) in various aspects by the Brick Agricultural Database, and then quantitatively scoring from the aspects of the spreading scope, severity and the propagation speed. By analyzing this index sequence, the Brick Agricultural Database defines an index of swine epidemics’ width less than 0.2 as a normal level, and an index of swine epidemics’ width greater than 0.25 as a serious situation. Therefore, in this study, the ISEW is used as a proxy variable to measure the severity of swine epidemics in China. So, it is assumed that the unit change in the ISEW will cause a unit change in slaughter, then the pig slaughter rate is: $S=\left(i_{0}/i\right)S_{0}$. The delay in pig slaughter is only a delay on the basis of the normal number of pigs slaughtered *S*. As affected by the influence of the swine epidemic and the delay in pig slaughter, the function of pig supply can be expressed as
(7)$$S=\frac{i_{0}}{i}\cdot \frac{t_{0}}{t}\cdot S_{0}$$
where *S*_0_ is the normal supply of live pigs before the outbreak of the epidemic, *i* is the actual ISEW at that time, and *i*_0_ is the normal level of the ISEW. Therefore, the total reduction in the pig supply is expressed as
(8)$$\Delta S=S_{0}\left(1-\frac{i_{0}}{i}\cdot \frac{1}{1+\ln\left(P\cdot c/VC\right)}\right)$$

It can be seen from Equation (8) that the larger the ISEW (*i*), the greater the price increase and the smaller the total supply of pigs. The equilibrium market price will rise continually because of the rapid recovery of consumers’ demand and the gradual expansion of the gap between supply and demand, resulting in a rise in pig prices, and then the breeding period of pigs will be further extended, further increasing the gap between supply and demand for pork. With the rise in pork prices, the “Matthew effect” of pig supply enters a vicious circle, causing violent fluctuations in the pig supply. Therefore, as the severity of the epidemics increases, the fluctuation in the pig supply presents nonlinear shapes and involves a threshold effect.

## 3. Empirical Analysis

The section of microscopic mechanism analysis explores the nonlinear impact of swine epidemics on the pig supply, while the section of empirical analysis is a proof of microscopic mechanism analysis and finds the threshold point on the impact of swine epidemics on the pig supply.

### 3.1. Variable Selection and Data Sources

This study selected data from January 2012 to June 2020 as the research sample, and the data of pig stock and slaughter were all from the Animal Husbandry Yearbook compiled by National Bureau of Statistics of China. The ISEW and other data are from the Bric Agricultural Database. The relevant data were processed through X-12 seasonal adjustment to filter out the influence of seasonal and cyclical factors. To eliminate the impact of heteroscedasticity on the pig stock and slaughter in China, the relevant data were log converted and some data related to prices were deflated by the consumer price index. This paper used the Eviews 10.0 software for data processing (Eviews, Irvine, CA, USA).

#### 3.1.1. Dependent Variables

Monthly pig stock (LNPIGSTOCK) and monthly pig slaughter (LNPIGSLAUGHTER): The pig supply mainly depends on the current pig stock in China, and the pig stock is the basis for pig slaughter. To minimize missing information, quadratic interpolation was used to convert the quarterly pig slaughter data into monthly data.

#### 3.1.2. Threshold Variables

The index of swine epidemics’ width (LNISEW): The variety of swine diseases and transmission routes lead to rapid changes in the epidemic, and it is difficult to obtain the relevant basic data. Therefore, the existing ISEW was selected as the variable to measure the epidemic. The index is a comprehensive index based on the evaluation of the outbreak scope, severity, and transmission speed of the swine disease in monitored regions, which mainly reflects changes in the outbreak range of the swine diseases. The diseases mainly include Blue ear disease, swine fever, acute diarrhea, and high fever. An ISEW less than 0.2 indicates a normal level, while an ISEW greater than 0.25 indicates a serious swine epidemic in China.

#### 3.1.3. Control Variable

In order to increase the robustness of the estimation results of the threshold model and control the impact of other factors such as consumer demand, breeding cost and breeding profit on pig supply in addition to the index of swine epidemics’ width, the control variables are set as follows:

Pig price (LNPIGPRICE): The number of agricultural products is greatly affected by the signal market price. Pig price is an important basis for price expectation during pig production, and producers will make production decisions according to their price expectations. Based on the availability of data, this study selected the existing factory price of pigs from the State Statistical Bureau in China. Breeding sow price (LNSOW): The stock of breeding sows is the beginning of the industrial chain of pigs. Producible sow stock determines the production of piglets and affects the pig supply, with a lag of 10 months, thus the price of sows has a great influence on the pig supply. Piglet price (LNPIGLET): Piglet price is important components of breeding cost of pig [26], it has an impact on pig supply by influencing producers’ decisions. Chicken price (LNCHICKEN): Chicken is the most important substitute for pork, and is a significant factor affecting the fluctuation in pig prices, and have a significant impact on the pig supply [9]. Based on the consistency of data, the average method was adopted to convert the existing weekly price of chicken into monthly data. Corn price (LNCORN): Corn is among the most important raw materials for feeding pigs, so the price of corn is among the main costs of swine production, and the corn price has a significant guiding and pulling effect on the price of pigs [27]. The corn price determines the cost of pig breeding and the ratio of pig to food (breeding profit), which has a significant impact on the pig supply. Feeding cost (LNFC): Feeding cost refers to the total cost of various feeds (including corn and soybean meal) required by pigs during the entire breeding period. Feed consumption accounts for the largest proportion of the pig breeding costs [28], which is up to 60%, and is an important factor affecting the pig supply. Breeding profit (PROFIT): Farmers pursue profit maximization under the condition of a market economy, and they will adjust the allocation of production resources according to the changes in breeding profit [29], which directly affects the supply of pigs.

The data that used in this article are shown in Appendix A. The descriptive statistics of all variables are shown in Table 1. It can be seen that the standard deviation of the LNISEW is larger than the general value, indicating that the fluctuation in swine epidemic is more intense.

### 3.2. Correlation Test of the Model

#### 3.2.1. Stationary Test

To avoid the occurrence of “spurious regression”, the most commonly used method is the augmented Dickey–Fuller (ADF) test, which is based on a residual for unit root test. The results of the test are shown in Table 2. It can be seen that the LNPIGSLAUGHTER and the LNISEW are stationary series, and the other variables are stationary series after processing of the first-order difference.

#### 3.2.2. Nonlinear Test

To exclude the influence of linear correlation components on the results of the nonlinear relationship test and obtain a more accurate conclusion, this study established the Auto Regressive (AR) model to test and filter linear correlation components for the pig stock and pig slaughter data, respectively. The results of the Lagrange Multiplier (LM) test showed that the series of pig slaughter reject the original hypothesis at the 1% significance level. It has auto-correlation, while the first-order difference series of the pig stock did not show first-order auto-correlation. The series of pig slaughter showed auto-correlation. Therefore, an AR (P) model with different lag orders should be used to fit the original series to eliminate the auto-correlation. After multiple tests, the residual series without auto-correlation was obtained using the AR (1) model to fit.

The Brock–Dechert–Scheinkman (BDS) test was performed on the first-order difference series of pig stock and the residual series of the AR (1) model of pig slaughter. To enhance the accuracy of the test results, Bootstrapping (10,000 times) was carried out to obtain the estimated *p*-value. The results showed that the Z-statistic of the BDS nonlinear statistical test obey the standard normal distribution. The *p*-values are 0 in several dimensions, and reject the null hypothesis at the 1% significance level, indicating that neither the residual series of pig slaughter nor the first-order difference series of pig stock are an independent and uniformly distributed (linear) series. Therefore, the results showed that there is a clustering property on fluctuations in pig stock and slaughter in China. Large fluctuations concentrate in some periods, while low fluctuations concentrate in other periods. This phenomenon can be analyzed using a threshold regression model.

#### 3.2.3. Model Specification

The threshold effect refers to the phenomenon that when one economic parameter reaches a specific critical value (threshold value), the other economic parameter suddenly turns to other developmental forms, which has a piecewise linear characteristic and shows nonlinear changes. The general form of the single-threshold regression model was set as follows:(9)$$y_{t}=\left(\alpha_{0}+\sum_{i=1}^{n}\alpha_{i}x_{i}\right)I\left(q_{i}≺\gamma \right)+\left(\beta_{0}+\sum_{i=1}^{n}\beta_{i}x_{i}\right)I\left(\gamma \leq q_{i}\right)+e_{i}$$

In the formula, *X_i_* is the explanatory variable, *I*(*q_i_*, *γ*) is an indicator function, denoting the threshold variable *q_i_* on the set of *γ*, *α_i_* and *β_i_* represent the regression coefficient, and *e_i_* is the random disturbance term. The conversion conditions of the first interval and the second interval are *q_i <_ γ*, *γ* ≤ *q_i_* respectively.

The threshold variable is the ISEW and is expressed as *q_ai_* in this study. Taking the logarithm of feeding cost (LNFC), pig price (LNPIGPRICE), producible sow price (LNSOW), piglet price (LNPIGLET), and breeding profit (PROFIT) as the control variables expressed by *α*_1_, *α*_2_, *α*_3_, *α*_4_, *α*_5_, the threshold regression equation of the pig stock is as follows:(10)$$y_{at}=\left(\alpha_{a}{}_{0}+\sum_{i=1}^{5}\alpha_{ai}a_{i}\right)I\left(q_{a}{}_{i}≺\gamma_{1}\right)+\left(\beta_{a}{}_{0}+\sum_{i=1}^{5}\beta_{a}{}_{i}a_{i}\right)I\left(\gamma_{1}\leq q_{a}{}_{i}\right)+e_{ai}$$

In the formula, the ISEW is the threshold variable, denoted by $q_{bi}$. Taking the logarithm of chicken price (LNCHICKEN), corn price (LNCORN), piglet price (LNPIGLET), breeding sow price (LNSOW), pig price (LNPIGPRICE), and breeding profit (PROFIT) as the control variables expressed by *b*_1_, *b*_2_, *b*_3_, *b*_4_, *b*_5_, *b*_6_, the threshold regression equation of pig slaughter is specified as follows:(11)$$y_{bt}=\left(\alpha_{b0}+\sum_{i=1}^{6}\alpha_{bi}b_{i}\right)I\left(q_{bi}≺\gamma_{2}\right)+\left(\beta_{b0}+\sum_{i=1}^{6}\beta_{bi}b_{i}\right)I\left(\gamma_{2}\leq q_{bi}\right)+e_{bi}$$

The alternative variables *A_i_*, *A_i_*(*γ*) and *B_i_*, *B_i_*(*γ*) are selected to replace Equations (10) and (11), *A_i_*, *A_i_*(*γ*) and *B_i_*, *B_i_*(*γ*) are, respectively, defined as:$$A_{i}=\left(1,a_{1},a_{2},a_{3},a_{4},a_{5}\right),\;A_{i}\left(\gamma \right)=\left(A_{i}^{\prime }I\left(q_{ai}≺\gamma_{1}\right)\right)A_{i}^{\prime }I\left(\gamma_{1}\leq q_{ai}\right),$$
$$B_{i}=\left(1,b_{1},b_{2},b_{3},b_{4},b_{5},b_{6}\right),\;B_{i}\left(\gamma \right)=\left(B_{i}^{\prime }I\left(q_{bi}≺\gamma_{2}\right)\right)B_{i}^{\prime }I\left(\gamma_{2}\leq q_{bi}\right).$$

Therefore, Equations (10) and (11) can also be expressed as follows:(12)$$y_{at}=A_{i}{\left(\gamma_{1}\right)}^{\prime }\theta_{a}+e_{a}{}_{i},\;\mathrm{where}\;\theta_{a}={\left(\alpha_{a}{}^{\prime }\beta_{a}{}^{\prime }\right)}^{\prime }$$
(13)$$y_{bt}=B_{i}{\left(\gamma_{2}\right)}^{\prime }\theta_{b}+e_{bi},\;\mathrm{where}\;\theta_{b}={\left(\alpha_{b}{}^{\prime }\beta_{b}{}^{\prime }\right)}^{\prime }$$

When the regression error is independent and identically distributed, the statistical method of maximum likelihood estimation was used to fit the regression. Considering that the threshold regression equations in this study are nonlinear and discontinuous, the sequential conditional likelihood estimation method was used for parameter estimation. In the case that the value of *γ*_1_ and *γ*_2_ are given, *θ**_a_* and *θ**_b_* were defined separately as follows:(14)$${\hat{\theta }}_{a}\left(\gamma_{1}\right)={\left(\sum_{i=1}^{5}A_{i}\left(\gamma_{1}\right)A_{i}{\left(\gamma_{1}\right)}^{\prime }\right)}^{-1}\left(\sum_{i=1}^{5}A_{i}\left(\gamma_{1}\right)y_{at}\right)$$
(15)$${\hat{\theta }}_{b}\left(\gamma_{2}\right)={\left(\sum_{i=1}^{6}B_{i}\left(\gamma_{2}\right)B_{i}{\left(\gamma_{2}\right)}^{\prime }\right)}^{-1}\left(\sum_{i=1}^{6}B_{i}\left(\gamma_{2}\right)y_{bt}\right)$$

The residuals were defined as follows:$${\hat{e}}_{ai}=y_{at}-A_{i}{\left(\gamma_{1}\right)}^{\prime }{\hat{\theta }}_{a}\left(\gamma_{1}\right).{\hat{e}}_{bi}=y_{bt}-B_{i}{\left(\gamma_{2}\right)}^{\prime }{\hat{\theta }}_{b}\left(\gamma_{2}\right)$$

The residual sums of squares are shown by the following equations:(16)$$\sigma_{5}^{2}\left(\gamma_{1}\right)=\frac{1}{5}\sum_{i=1}^{5}e_{ai}{\left(\gamma_{1}\right)}^{2}$$
(17)$$\sigma_{6}^{2}\left(\gamma_{2}\right)=\frac{1}{6}\sum_{i=1}^{6}e_{bi}{\left(\gamma_{2}\right)}^{2}$$

Assuming that *γ*_1_ and *γ*_2_ are confined to a bounded set $\left[\bar{\gamma },\underline{\gamma }\right]$:(18)$${\hat{\gamma }}_{1}=\arg i\min\left(\sigma_{5}^{2}\left(\gamma_{1}\right)\right)$$
(19)$${\hat{\gamma }}_{2}=\arg i\min\left(\sigma_{6}^{2}\left(\gamma_{2}\right)\right)$$

If we specify *γ*_1_ and *γ*_2_ to the threshold variables *q_ai_* and *q_bi_* then *γ*_1_ and *γ*_2_ are calculated as follows:(20)$${\hat{\gamma }}_{1}=\arg i\min\left(\sigma_{5}^{2}\left(q_{ai}\right)\right)$$
(21)$${\hat{\gamma }}_{2}=\arg i\min\left(\sigma_{6}^{2}\left(q_{bi}\right)\right)$$

## 4. Results and Discussion

### 4.1. Results

As shown in Table 3, at a significance level of 1% and selected by Bai-perron Critical Value, a single threshold value is the optimal choice, which further verified the scientific quality of the selected model. When the logarithm of the ISEW over 7 months (LNISEW-7) was used as the threshold variable for the pig stock, the estimated parameter of threshold value was 1.3863. While the estimated parameter of threshold value was 1.1087 when the logarithm of the ISEW over 8 months (LNISEW-8) was used as the threshold variable of pig slaughter. The interval transition of the pig stock that was affected by the severity of swine epidemics was earlier than that of pig slaughter, while the fluctuation in pig slaughter affected by the severity of epidemics was higher than that of the pig stock. The specific impacts were as follows.

(1) When the ISEW was less than 0.25, the impact of swine epidemics on pig stock was not significant (*p* > 0.05), i.e., when the epidemic was below a serious level, the pig stock did not change markedly. However, when the ISEW was higher than 0.25, the negative impact of swine epidemics on the number of pigs slaughtered was significant (*p* < 0.01).

(2) When the ISEW was less than 0.33, the positive effect of swine epidemics on the number of pigs slaughtered was significant (*p* < 0.01), while epidemics had a significant negative impact on the pig slaughter rate when the ISEW was higher than 0.33 (*p* < 0.01).

### 4.2. Discussion

(1) The contribution of this paper. Compared with the existing literature, this manuscript pays more attention to the nonlinear relationship between swine epidemics and the fluctuation in pig supply. This paper reveals the microscopic mechanism of the nonlinear change in pig supply in the outbreak phase of swine epidemics, which will enrich the relevant theoretical research on the nonlinear fluctuation in pig supply and finds the threshold effect of swine epidemics (using the index of swine epidemics width as the proxy) on the supply of pigs in China by identifying the threshold values that caused the drastic fluctuation in pig supply, that is, swine epidemics can cause nonlinear fluctuations in pig supply, so as to provide suggestions for the establishment of early warning mechanism of pig supply. This finding is of substance in the context of rapidly developing of China’s hog industry.

(2) The limitations of this paper. Firstly, most of the research on swine epidemics focused on natural science or the impact of swine epidemics on pork prices. There are few studies available in this paper to explore the impact of swine epidemics on the fluctuation in pig supply, so there are deficiencies in summarizing the research advance around the topic of the impact of swine epidemics on pig supply. Secondly, the objective variables to measure the severity of swine epidemics include the mortality of pigs, the culling volume of sick pigs, public opinion on the epidemics, and the index of swine epidemics’ width. However, due to the difficulty of data acquisition, this paper only uses the index of swine epidemics’ width as the proxy variable of swine epidemics, which may be insufficient.

## 5. Conclusions and Suggestions

### 5.1. Conclusions

This study constructed a production function and a profit function of pigs based on the law of diminishing marginal returns and the equilibrium conditions of profit maximization. According to the theory analysis, the influence of different degrees of swine epidemics on China’s pig supply is also different, and we proposed that swine epidemics have a nonlinear effect on the fluctuation in China’s pig supply. Then, a threshold regression model was used to analyze the threshold effect of different severities of swine epidemics on the pig stock and slaughter rate in China. The main conclusions were as follows: 

(1) In the threshold regression results of the ISEW (lag = −7) on the pig stock, there was a threshold value of 1.3863 that divided the sample into two intervals. When the LNISEW-7 was less than 1.3863, i.e., when the actual ISEW was less than 0.25 (the epidemic was below the serious level), the impact of the epidemic on the pig stock is not significant. Meanwhile, the impact of the epidemic on pig stock is negative and significant at the 1% level when the LNISEW-7 was greater than or equal to 1.3863, i.e., when the actual ISEW is higher than 0.25 (the epidemic is above the serious level), and the decrease coefficient was 0.016351.

(2) In the threshold regression results of the ISEW (lag = 8) for pig slaughter, there was a threshold value of 1.1087 that divided the sample into two intervals. When the LNISEW-8 was less than 1.1087, i.e., when the actual ISEW was less than 0.33, swine epidemics had a positive effect on pig slaughter that was significant at the 1% level, with a coefficient of 0.074382. When the LNISEW-7 was greater than or equal to 1.1087 (ISEW > 0.33), the impact of swine epidemics on pig slaughter was negative and significant at the 1% level with a decreased coefficient of 0.644362.

The above conclusions indicated that the current supply system of pigs in China can maintain the basic stability of operation in response to small-scale epidemics. Nevertheless, in the face of serious epidemics, the negative impact on the pig supply is significant and becomes stronger because of the threshold effect of swine epidemics on the pig supply in China. This transformation and enhancement reflect the fact that the supply systems of agribusinesses that breed pigs and produce pork in China are affected by multiple factors, such as policy regulation and control, marketing mechanisms, technology research and development, and lack robustness.

### 5.2. Policy Suggestions

Strengthen the construction of the monitoring system in the pig industry and improve the efficiency of intervention mechanisms in the pig supply. Firstly, refer to the different threshold ISEW for regime switching to promote the construction of an early warning system, which could ensure government departments acquire information and launch schemes for epidemic prevention and control promptly and accurately, and implement prophylactic measures at any time, to avoid the spread of the epidemic. Secondly, improve the pork reservation and delivery system to implement accurate and timely regulation and improve the efficiency of reserved pork release to the market. Thirdly, a standard slaughtered weight institution should be implemented after the outbreak of swine epidemic. The departments of husbandry management should set a series of standard slaughtered weights for different pig varieties and introduce a ladder price for overweight pigs to guide slaughter at the proper time according to the standard weight. These measures would alleviate the abnormal supply interruption and the continuous high price of pork.

Reinforce the moderating effect of market mechanisms and develop tools to regulate the situation in the pig market. Firstly, implement the construction of an insurance system for pig feeding and develop types of insurance to adjust the insurance conditions, period, and compensation of farmers and agribusinesses without delay according to the effects of different degrees of epidemics, to reduces losses and stabilize returns for producers in infected areas. In the meantime, reduce the expectations of price increases for producers in non-infected areas during the outbreak of swine epidemics. Secondly, the construction of live pig futures market and management systems should be improved to promote the standardization of assigning products and the adaptation of the futures system to the structure of the pig supply in China. The hedging and price discovery functions of pig futures should be used to help breeders avoid price risks after the outbreak of swine epidemics and improve the efficiency of resource allocation in the pig industry. Thirdly, the rational consumption of pork substitutes should be further promoted, which would enhance the productivity of other proteins with a short growth cycle, such as chicken and beef, which would inhibit the excessive rise in pork prices, and also affect the price expectations of pig breeders, adjusting the production decisions made by farmers indirectly.

Encourage the development and popularization of technologies that prevent swine epidemics and enhance the sustainability of the pig supply. Firstly, increase investment in vaccine research and development, and enhance the R&D ability and supply capacity of products that can prevent the outbreak of swine epidemics. Secondly, enhance the cultivation, promotion, and genetic modification of favorable varieties and increase the productivity and the contribution of scientific and technological progress in China’s pig industry. Thirdly, pig breeding systems should be strengthened to ensure the reasonable structure of stocking pigs for propagation, commercial pigs, and piglets, which would increase the production capacity and form an effective system of pig production. Fourthly, the pattern of double-circulating between international and domestic for pork supply should be constructed quickly to enhance the robustness and elasticity of the pork supply system in China and alleviate the influence of epidemic diseases in swine.

## Figures and Tables

**Figure 1 animals-12-02595-f001:**
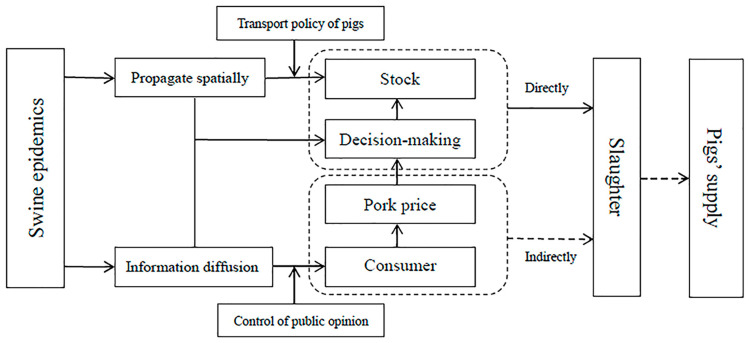
The conduction path of the impact of swine epidemics on the pig supply.

**Figure 2 animals-12-02595-f002:**
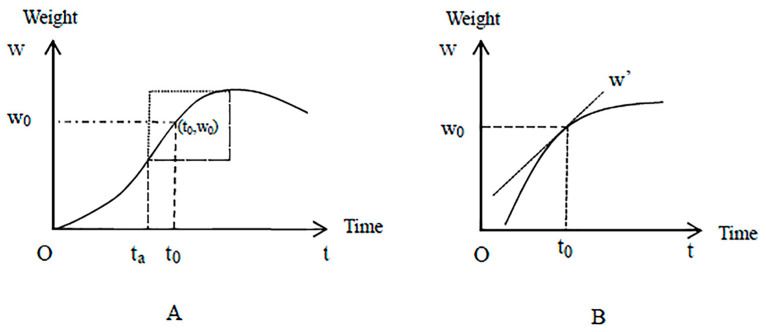
The graph of pig weight versus breeding time. (**A**) The graph of pig weight versus breeding time. (**B**) The graph of the relationship between breeding time (after t_a_) and pig weight.

**Table 1 animals-12-02595-t001:** Descriptive statistical characteristics of the variables.

Variables	Mean	Standard Deviation	Min	Max
LNPIGSTOCK	10.503	0.233	9.909	10.763
LNPIGSLAUGHTER	4.223	0.057	4.033	4.269
LNISEW	−1.358	0.405	−2.526	−0.301
LNPIGPRICE	1.392	0.094	1.267	1.677
LNSOW	0.032	0.996	−1.998	1.325
LNPIGLET	2.589	0.153	2.371	3.074
LNCHICKEN	1.244	0.029	1.189	1.345
LNCORN	3.280	0.073	3.153	3.382
LNFC	2.812	0.075	2.691	2.917
PROFIT	328.100	487.632	−250.649	2224.719

**Table 2 animals-12-02595-t002:** The results of the stationary test (ADF test).

Variables	The Original Series		The First-Order Difference Series	
	ADF Statistic	1% Value Critical Level	ADF Statistic	1% Value Critical Level
LNPIGSTOCK	−2.776863	−4.054393	−3.408006	−2.588772
LNPIGSLAUGHTER	−5.475513	−4.060874	NA	NA
LNISEW	−4.837696	−3.501445	NA	NA
LNPIGPRICE	0.604110	−2.588772	−6.218526	−2.588772
LNFC	0.212341	−2.588772	−7.549460	−2.588772
LNSOW	−3.198424	−4.054393	−3.205999	−2.588772
LNPIGLET	0.932634	−2.589020	−6.289232	−2.589020
PROFIT	−3.164882	−4.054393	−6.215998	−2.588772
LNCHICKEN	−2.390845	−3.499167	−8.687411	−2.589020
LNCORN	−0.285288	−2.588772	−7.141257	−2.588772

NA, not available.

**Table 3 animals-12-02595-t003:** Threshold regression results.

Variable	Threshold Variable: LNISEW-7 To: Pig Stock		Threshold Variable: LNISEW-8 To: Pig Slaughter	
	The First Interval LNISEW-7 < −1.3863	The Second Interval −1.3863 ≤ LNISEW-7	The First Interval LNISEW-8 < −1.1087	The Second Interval −1.1087 ≤ LNISEW-8
LNISEW-7	−0.008418	−0.016351 ***	NA	NA
LNISEW-8	NA	NA	0.074382 ***	−0.644362 ***
LNFC	0.027045	0.013274	NA	NA
LNPIGPRICE	0.117452 ***	0.056755 ***	−0.296232 *	−0.899547 *
LNSOW	0.091570 ***	0.130785 ***	−1.253687 ***	−1.029459
LNPIGLET	−0.030654	−0.030949 ***	−0.270312 ***	−0.252805
PROFIT	−3.96 × 10^−5^ **	−1.32 × 10^−5^ *	0.000309 ***	0.000248 *
LNCHICKEN	NA	NA	0.079132	−0.209705
LNCORN	NA	NA	0.389657 **	−0.048691
R2	0.91896		0.92268	

Note: *** *p* < 0.01, ** *p* < 0.05 and * *p* < 0.1. NA, not available.

## Data Availability

The data used in this study are available from the corresponding author on request.

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
