# Peer review of "The Threshold Effect of Swine Epidemics on the Pig Supply in China"

_animals, 2022, doi:10.3390/ani12192595_

Round 1

Reviewer 1 Report

This manuscript has certain academic value and given some meaningful results, and there are some advice as following.

1. There are many new studies about swine epidemics influence on supply, demand and pork prices,especially after the outbreak of ASF in recent two years. So please update the references related to the main topic of the manuscript.

2. What about the research advance around the topic of the paper? The introduction should be added and the advance should be enriched, which would be appreciated.

3. The functional path of the impact of swine epidemic on pig supply belongs to mechanism, so the part should be relocated to the part about the mechanism.

4. The marginal weight gain equation has been given, which might be the shining of this manuscript, but the following empirical model has given another differently. So the relevance of theoretical analysis and empirical study before and after the article is greatly reduced. Better coherence and interpretation are necessary to bridge the gap between the two.

5.Because of the former mentioned about the lack of literature research, the contribution of the manuscript, as well as the difference with the results of the existing literature, need be further discussed at the end of its result part. 

Reviewer 2 Report

The idea of the paper is good, but I was wondering if this journal is the right place to publish the paper. Although the authors elaborated on the formulas used but it was not indicated whether the data (from January 2012 - June 2022) was made available as supplementary data.  

Secondly, the language needs to be improved, and the appropriate use of words must be ensured to avoid wrong interpretations. Acronyms are also used, or comments are introduced midway into a sentence but never explained, like ono-linear, Matthew effect, etc. Similarly. The introduction has an objective and an aim that are not related, and one was never discussed in the manuscript.

The authors also used words or figures without referencing the source; for example, "..values less than 0.2 indicates a normal le affects vel, while an ISEW value greater than 0.25 indicates a serious swine epidemic in China." What is the source of these figures, or are they from this study?

The authors also need to explain variables like a piglet, chicken corn, and feeding costs affects the model.

Lastly, the conclusion is longer than the result/Discussion, any reason for that? 

Round 2

Reviewer 1 Report

The author has revised and improved the suggestions, which is worthy of recognition.

There are three tips for the author.

1. As for Figure 1, the slaughter quantity determines the supply quantity, but the slaughter quantity mainly depends on two aspects. One is the number of live pigs on hand after the pig epidemic (only the highly infectious epidemic of African swine fever will restrict the circulation, but not all), which comes from farmers; Second, consumers' awareness of animal epidemics causes changes in consumption behavior, which then acts on the market, forming price pressure, transmitting the signal of "weakening demand", affecting slaughter volume, which is an indirect impact, and this comes from the consumer side.

The existing description in Figure 1 does not accurately reflect the author's intention, so please further modify and improve it as appropriate.

2.The discussion has been added. But the contribution of this article should be summarized briefly, and the differences between the existing literature and the findings of the manuscript should be given, but not only the contribution. As well, pay attention attention to the nonlinear relationship…… might be not its contribution, if make it sure might be one of the contributions.

3. Please pay attention to the application of punctuation between the First, Second and Third and so on, especially the semicolon (;).  

Reviewer 2 Report

I am very OK with the improvement made by the authors

Author Response

Thank you for your valuable comments and recognition of our articles, they mean a lot to us!